# Angiotensin II Influences Pre-mRNA Splicing Regulation by Enhancing *RBM20* Transcription Through Activation of the MAPK/ELK1 Signaling Pathway

**DOI:** 10.3390/ijms20205059

**Published:** 2019-10-12

**Authors:** Hanfang Cai, Chaoqun Zhu, Zhilong Chen, Rexiati Maimaiti, Mingming Sun, Richard J. McCormick, Xianyong Lan, Hong Chen, Wei Guo

**Affiliations:** 1Department of Animal Science, University of Wyoming, Laramie, WY 82071, USA; caihanfang.cool@163.com (H.C.); czhu1@uwyo.edu (C.Z.); zhilongchen@sina.com (Z.C.); rmaimait@uwyo.edu (R.M.); msun2@uwyo.edu (M.S.); RMcCrmck@uwyo.edu (R.J.M.); 2College of Animal Science and Technology, Northwest A&F University, Yangling, Shaanxi 712100, China; lan342@126.com

**Keywords:** RBM20, hormones, insulin, T3, angiotensin II, pre-mRNA splicing

## Abstract

RNA binding motif 20 (RBM20) is a key regulator of pre-mRNA splicing of titin and other genes that are associated with cardiac diseases. Hormones, like insulin, triiodothyronine (T3), and angiotensin II (Ang II), can regulate gene-splicing through RBM20, but the detailed mechanism remains unclear. This study was aimed at investigating the signaling mechanism by which hormones regulate pre-mRNA splicing through RBM20. We first examined the role of RBM20 in Z-, I-, and M-band titin splicing at different ages in wild type (WT) and *RBM20* knockout (KO) rats using RT-PCR; we found that RBM20 is the predominant regulator of I-band titin splicing at all ages. Then we treated rats with propylthiouracil (PTU), T3, streptozotocin (STZ), and Ang II and evaluated the impact of these hormones on the splicing of titin, LIM domain binding 3 (*Ldb3*), calcium/calmodulin-dependent protein kinase II gamma (*Camk2g*), and triadin (*Trdn*). We determined the activation of mitogen-activated protein kinase (MAPK) signaling in primary cardiomyocytes treated with insulin, T3, and Ang II using western blotting; MAPK signaling was activated and *RBM20* expression increased after treatment. Two downstream transcriptional factors c-jun and ETS Transcription Factor (ELK1) can bind the promoter of *RBM20*. A dual-luciferase activity assay revealed that Ang II, but not insulin and T3, can trigger ELK1 and thus promote transcription of *RBM20*. This study revealed that Ang II can trigger ELK1 through activation of MAPK signaling by enhancing *RBM20* expression which regulates pre-mRNA splicing. Our study provides a potential therapeutic target for the treatment of cardiac diseases in RBM20-mediated pre-mRNA splicing.

## 1. Introduction

Pre-mRNA splicing or alternative splicing is one of the most important mechanisms in post-transcriptional regulation. Pre-mRNA splicing promotes protein diversity in an organism because multiple, mature mRNAs containing different exons can be generated from a single gene and translated to distinct but functionally similar proteins [1,2]. Aberrant pre-mRNA splicing may result in developmental problems as well as human diseases [3,4]. Although pre-mRNA splicing has been extensively studied and splicing mechanisms of some genes have been well characterized, the splicing mechanisms of numerous genes are still poorly understood given the complexity of splicing regulation of each individual gene. Titin is one of these genes. It consists of 363 exons encoding the largest protein known to-date. Titin is responsible for the passive elasticity of heart muscle [5]. Studies have shown that titin undergoes numerous splicing events producing millions of isoforms of differing sizes [6,7,8]. However, the mechanism governing titin splicing had not been deciphered until a new splicing factor, RNA binding motif 20 (RBM20) was identified [9].

Rbm20 is a muscle-specific splicing factor containing two conventional domains called the RNA recognition motif (RRM) and the serine-arginine (SR) domain. Thus, it belongs to the family of SR proteins that predominantly regulate pre-mRNA splicing of the titin gene as well as 30 other genes including the calcium/calmodulin-dependent protein kinase II gamma (*Camk2g*), LIM domain binding 3 (*Ldb3*), and triadin (*Trdn*) [9]. Titin splicing can occur in three regions at the Z-band, I-band, and M-band [10,11,12,13,14]. Previous studies report that RBM20 regulation of alternative splicing alters titin size [9,12,15]. It is unclear, however, which titin regions or exons are regulated by RBM20. Studies have also shown that titin splicing can be regulated by hormonal activation (insulin, thyroid hormone, and angiotensin II (Ang II)) of the PI3k/Akt signaling pathway which up-regulates *RBM20* expression and titin splicing [16,17]. Thus, hormone-regulated titin splicing is RBM20 dependent [18,19]. However, whether other signaling pathways involve RBM20-mediated pre-mRNA splicing remains unknown.

This study was therefore designed to examine the role of RBM20 in the regulation of Z-, I-, and M-band titin splicing as well as the role of hormone (insulin, triiodothyronine (T3), and Ang II)-mediated pre-mRNA splicing. Signaling mechanisms by which RBM20 regulates pre-mRNA splicing stimulated by hormones were explored as well.

## 2. Results

### 2.1. Splicing Regulation of RBM20 in Z-, I-, and M-Band Titin at Different Ages

The titin gene contains 363 exons of which approximately 200 are alternatively spliced, producing very complicated splicing patterns [6,11]. Previous studies have shown that titin splicing is regulated developmentally in a tissue-specific manner [11,13,20,21,22,23]. Exon usage in titin splicing events occurs mainly in three regions—Z-band, I-band, and M-band [18,19]. A new splicing factor, RBM20, was recently identified as a major factor of titin splicing [9]. However, how RBM20 modulates exon splicing in these regions with developmental age remains unknown. Using RT-PCR, we determined splicing patterns in the Z-, I-, and M-band regions in the heart at day 1, 3- and 6-months with or without RBM20. Primers spanning exons 7–10 and exons 10–14 were used for detection of Z-band splicing, exons 54–56 and 66–68 for the I-band, and exons 361–363 for the M-band. In the Z-band, no changes were observed in exons 7–10 between wild type (WT) and the *RBM20* knockout (KO) at different ages, and only one variant (v1) was present in both WT and KO heart tissue (Figure 1A). There were four variants (v2–5) detected in exons 10–14. Quantification of the v2, 3, and 5 revealed that their expression level did not differ by age in the KO heart, however, v4 expression increased significantly in an adult heart compared to a neonatal heart (Figure 1A,B). Most interestingly, we observed that there were no splicing differences (i.e., no variants) between WT and KO at day 1, but in an adult heart, v3 tends to increase in the KO compared to the WT (Figure 1A,B). In the I-band, exons 54–56 produced v6 and v7, and exons 66–68 produced v8 and v9 in the WT heart; these variants were persistently expressed in the WT heart from neonate to adult rats, but their ratios (v6:v7 and v8:v9) changed during development. Furthermore, we observed that only the smaller variants, v7 and v9, were present in both KO neonate and adult hearts (Figure 1A,B). Lastly, in the M-band, with exons 361–363, only one variant (v10) was expressed in both WT and KO across all ages (Figure 1A). While it is known that RBM20 is a splicing factor, it cannot be assumed that all changes in splicing patterns seen in *RBM20* KOs result from loss of RBM20 activity at the exons being examined, and loss of RBM20 could have indirect effects on splice variant stability or non-regulated loss of splicing fidelity. In summary, these results suggest that splicing patterns were not significantly changed by RBM20 in either Z- or M-bands, but it remarkably regulates I-band titin and the regulation changes dynamically with development in the WT heart, but not in the KO heart. Notably, splicing pattern in Z-band has no change which includes all four variants (V2–5) with development, but expression level of v4 differs at different ages. The possibilities causing this could be either a different role of RBM20 in different regions of the titin mRNA at different ages, or a variable sensitivity of the titin mRNA to regulators of splicing fidelity in the absence of RBM20 and prone to splicing errors, or varied stabilities of alternative splice products in the absence of RBM20.

### 2.2. Impact of Hormones on the Regulation of Z-, I-, and M-Band Titin Splicing with and without RBM20

Regulation of RBM20-facilitated pre-mRNA titin splicing is not completely understood. Previous reports indicate that hormones, such as thyroid hormone, Ang II [17,18], and insulin [16,19] regulate titin size changes through RBM20, but the exons where these alterations occur have not been identified. Here, we determined the role of hormones in Z-, I-, and M-band titin splicing using *RBM20* KO rats. These rats were treated with propylthiouracil (PTU)—a drug that inhibits thyroid hormone secretion, T3, and streptozotocin (STZ)—a drug that inhibits insulin secretion at 6-month old, and Ang II at 3-month old. After treatment, heart tissues were collected and RNA was isolated and purified from these tissues. Splicing patterns of Z-, I-, and M-band titin were examined using the primers mentioned above. WT rats were used as control. In the Z-band region, no treatments influenced the splicing pattern of exons 7–10 in either WT or KO hearts, however T3, STZ, and Ang II treatments produced significant differences in variant distribution between treatment and control (*p* < 0.05) in exons 10–14 in the WT heart. Treatment had no effect on splicing pattern in KO hearts (Figure 2A–D).

In the I-band, the ratios of larger to smaller variants (v6:v7 and v8:v9) in exons 54–56 and exons 66–68, respectively, demonstrated no significant difference with T3 and STZ treatments in WT compared to control, but the ratios were decreased significantly by PTU treatment and increased by Ang II treatment (*p* < 0.05). Only one variant, either v7 or v9, was observed in the KO heart, and the level of expression and splicing pattern was unchanged in both treated and untreated groups (Figure 2A–D). In the M-band (exons 361–363), only one variant (v10) was detected and its expression level, as well as its splicing pattern, were the same in both WT and KO regardless of treatments. To summarize, although hormone treatments can alter the expression level of some variants in Z- and I-band titin in the WT group, splicing pattern was not altered. Splicing pattern differences between WT and KO hearts were only observed in the I-band.

### 2.3. Hormone Impact on the Splicing of Other Substrates

In addition to titin, RBM20 can also regulate splicing of 30 more genes in the heart that are associated with the development of heart failure [9,15]. Here, we tested the impact of hormones on the splicing of three other targets of RBM20 including the *Camk2g*, *Ldb3*, and *Trdn*. As reported previously, all of these genes have two isoforms produced by alternative splicing [9]. In the control group, *Ldb3* has a higher ratio of larger variant (v11) to smaller variant (v12) in the WT heart compared to the KO heart. In the PTU treatment group, the v11:v12 increased in the KO, but not in the WT; in the STZ treated group, the v1:v2 was not changed in either the WT or the KO when compared to controls; however, in the T3 and Ang II treatment groups, the v11:v12 was unchanged in the KO but deceased in the WT when compared to the control groups (Figure 3A–D). *Camk2g* has a higher ratio of larger (v13) to smaller variant (v14) in the KO than in the WT heart in absence of treatment. In the PTU treatment group, the v13:v14 was unchanged in both WT and KO when compared to their respective controls. In the T3 and STZ treated groups, the v13:v14 was significantly decreased in the WT when compared to the WT control, but no ratio change was observed in the KO group when compared to the control group. With Ang II treatment, the v13:v14 in the WT and KO was decreased compared to the controls (Figure 3A–D). Trdn demonstrated a higher ratio of the larger (v15) to the smaller variant (v16) in untreated KO than in untreated WT hearts (control groups). With PTU treatment, the v15:v16 was significantly reduced in the WT and KO when compared to control groups (*p* < 0.05). With T3 and STZ treatment, the v15:v16 was reduced in the WT when compared to the control, but increased in the KO when compared to the control. With Ang II treatment, no ratio change was observed in the WT, but the v15:v16 was reduced in the KO heart compared to the control (Figure 3A–D). In summary, hormones can regulate gene splicing in *Ldb3*, *Camk2g*, and *Trdn* in both WT and KO heart tissues, but given the absence of RBM20 in KO hearts, it would appear that splicing is not RBM20 dependent, suggesting that regulation of these genes could be orchestrated by other factors.

### 2.4. Activation of MAPK Signaling by Insulin, Thyroid Hormone, and Ang II in Cultured Neonatal Cardiomyocytes

To determine the signaling mechanism of RBM20-mediated hormone regulation, we treated cultured primary cardiomyocytes with insulin, T3, and Ang II and examined activation of the MAPK signaling pathway. Our previous studies have shown that titin splicing can be regulated by insulin and T3 via the PI3K/AKT/mTOR signaling pathway in an RBM20-dependent manner [18,19]. Activation of the PI3K/AKT/mTOR kinase axis can induce *RBM20* expression, and thus modulate titin isoform transition [19]. However, these hormones can also activate the MAPK signaling pathway, so in this study, we assessed the activation of MAPK signaling in primary cardiomyocytes treated with these hormones. After treatment for 48 h with insulin, T3, and Ang II, neonatal cardiomyocytes were harvested and RNA was prepared to evaluate the *RBM20* transcription level with real-time qPCR. Insulin, T3, and Ang II significantly increased the transcriptional level of *RBM20* mRNA (*p* < 0.05) (Figure 4A). Previous reports showed that insulin, T3, and Ang II can activate MAPK signaling, leading to increased gene transcription [24,25,26]. Therefore, we also prepared protein lysates from the cardiomyocytes 5–10 min after treatment and examined the MAPK signaling hallmark proteins p38, ERK1/2, and JNK by western blotting. Results revealed that phosphorylation levels of p38, ERK1/2, and JNK were significantly higher in the cardiomyocytes treated with insulin, T3, and Ang II than in the untreated cardiomyocytes (*p* < 0.05) (Figure 4B–E). These results suggest that increased *RBM20* mRNA could be associated with the activated MAPK signaling pathway.

### 2.5. MAPK Signaling Activation Enhances RBM20 Transcription in Primary Neonatal Cardiomyocyte Cultures

To test the hypothesis that activation of MAPK signaling promotes *RBM20* transcription, we identified MAPK signaling-activated transcriptional factors that can potentially bind the promoter region of the *RBM20* gene using online tools PROMO 3, JASPAR, and MatInspector. We evaluated the role of these transcriptional factors in the *RBM20* transcription with an in vitro luciferase reporter assay. We found two downstream transcriptional factors of the MAPK signaling pathway: c-jun with one predicted binding site on the *RBM20* promoter (Figure 5A), and *ELK1* with multiple binding sites on the *RBM20* promoter (Figure 5B). Using a dual-luciferase activity assay, we detected the transcriptional activity of the c-jun and ELK1 binding sites on the *RBM20* promoter in H9C2 cells treated with insulin, T3, and Ang II. All three external stimulators significantly enhanced the relative luciferase activity of the *RBM20* promoter (*p* < 0.05) when compared to untreated controls (Figure 5C). To test whether transcriptional factors c-jun and ELK1 are critical for hormone-promoted transcription of *RBM20*, we introduced site-directed deletions of the binding sites of c-jun (binding site 1, BS1) and ELK1 (binding site 2, BS2). The dual-luciferase activity assay was performed in H9C2 cells and the results indicated that the deletion of BS1 did not significantly eliminate the insulin-, T3-, and Ang II-mediated activity of the *RBM20* promoter (*p* > 0.05) (Figure 5D). However, the deletion of BS2 diminished the effect of Ang II on the *RBM20* promoter (*p* < 0.05), but not insulin and T3 (*p* > 0.05) (Figure 5D). These results suggest that insulin, T3, and Ang II activate MAPK signaling and enable transcription of *RBM20*, but among tested hormones, only Ang II can activate transcriptional factor ELK1-bound promoter of *RBM20*. Insulin and T3 may regulate *RBM20* transcription through other mechanisms that have not been investigated in this study. However, further study is warranted.

## 3. Discussion

In the present study, we first examined how external stimuli affect pre-mRNA splicing of Z-, I-, and M-band titin in an *RBM20*-dependent manner. Consistent with previous studies [18,19], insulin, T3, and Ang II can regulate titin isoform switching in the presence of *RBM20* in heart tissue. These stimuli also increased the mRNA expression of *RBM20* through increased transcriptional activity. This study reports for the first time that Ang II can activate the MAPK signaling pathway, and thus promote transcriptional factor ELK1 activity for the promoter region binding of *RBM20*, leading to elevated *RBM20* transcription and expression.

Alternative splicing is an important post-transcriptional process in the regulation of gene expression [1,2]. Abnormal splicing of sarcomeric and ion channel genes can ultimately alter the normal internal architecture and homeostasis of the heart that can promote the heart remodeling and may ultimately lead to heart failure [10,27,28,29,30]. Titin-based passive stiffness is primarily modulated by isoform switching resulting from alternative splicing. Switching to the larger titin N2BA isoform will result in more compliant ventricular walls, while switching to the smaller N2B titin isoform may lead to less compliant ventricles [31]. Titin isoform transitions have occurred in a number of cardiac diseases, such as dilated cardiomyopathy, diabetic cardiomyopathy, ischemic heart failure, and hypertrophic cardiomyopathy [32]. These studies suggest that adjusting pre-mRNA splicing could be a therapeutic strategy for the treatment of human heart diseases. This study focused mainly on signaling regulation of RBM20 in Z-, I-, and M-band titin splicing and external stimuli-induced splicing. The smaller N2B isoform predominates in the hearts of rodents, large animals, and human beings. The N2B isoform results primarily from the exon exclusion from exons 50 to 219 [12]. Furthermore, exon skipping in titin splicing events commonly occurs from exons 50 to 101. These regions undergoing frequent splicing are located in the titin I-band [6,11]. Therefore, we designed primers (exons 54–56 and exons 66–68) to randomly sample the I-band titin splicing regions regulated by RBM20. In the Z- and M-band, splicing events occur from exons 10 to 14 and exons 361 to 363, respectively. Primers spanning exons 10–14 and exons 361–363 were designed to represent Z- and M-band titin. Previous studies reported that titin splicing is regulated developmentally [21,22,33]. This study indicates that the Z-band variant (v4) experiences developmental changes, but not other variants (v2, 3, and 5). The M-band variant does not undergo developmental changes. I-band titin variants are drastically regulated developmentally. We also found that RBM20 is not a major regulator for Z- and M-band titin splicing which is consistent with our previous study [14]. These results help us to further understand splicing regulation of RBM20 in exons located in different regions of titin and provide new information on which titin exons can be altered via RBM20 resulting in variable titin sizes. These observations suggest a plausible approach for the treatment of heart diseases.

Published work has shown that insulin, thyroid hormone (T3), and Ang II can regulate titin isoform transition through the PI3K/AKT signaling pathway [16,17,18,19], and the PI3K/AKT/mTOR kinase axis regulates titin isoform transition in an *RBM20*-dependent manner [18,19]. However, titin isoform transition regulated through the PI3K/AKT/mTOR kinase axis is subtle. In other words, the change is not sufficient for altering ventricular stiffness. This suggests further studies are required to identify an efficient mechanism regulating titin isoform transition. The MAPK signaling pathway consists of a chain of proteins in the cell that transduces a signal from a receptor on the surface of the cell to the DNA in the nucleus of the cell [34]. Insulin, T3, and Ang II can activate the MAPK signaling pathway in various cell types [24,25,26], such activation is closely associated with cardiac diseases [35,36,37]. Furthermore, studies indicate that the MAPK signaling pathway participates in gene splicing [38]. Hence, we hypothesize that the MAPK signaling may also regulate *RBM20* expression, and thus titin splicing. In this study, we focused on investigating the regulation of *RBM20* transcriptional activation through the MAPK signaling. Insulin, thyroid hormone, and Ang II have been shown to trans-activate gene transcription and mRNA expression. For example, insulin regulates sterol regulatory-element-binding protein-1c transcriptional activity in rat hepatocytes [39]. After T3 treatment in neonatal rat ventricular myocytes, brain natriuretic peptide gene mRNA level and transcriptional activity were increased 3-fold and 3–5-fold, respectively [40]. Ang II can induce interleukin-6 transcription in vascular smooth muscle cells through pleiotropic activation of nuclear factor-κB transcriptional factors [41]. Parallel to these results, we observed that insulin, T3, and Ang II can regulate *RBM20* transcriptional activity and expression.

It is well known that the activated MAPK signaling pathway phosphorylates ELK1 that in turn leads to trans-activation of gene expression via the serum responsive element (SRE) on the promoter region [42]. Our data indicate that ELK1 can bind to the promoter region of *RBM20*. Interestingly, Ang II can promote the binding of ELK1 to the promoter of *RBM20* and enhance the transcriptional activity of *RBM20*. Previous studies indicated that hormones can induce phosphorylation of ELK1 through the MAPK signaling pathway and thereby control gene transcription and expression [43,44]. However, the present results suggest that insulin and T3 do not regulate *RBM20* transcriptional activity via ELK1. Our previous studies showed that insulin and T3 can regulate *RBM20* expression via the PI3K/Akt/mTOR signaling pathway [18,19]. These results suggest that external stimuli regulate the expression level of the same gene through different pathways. In addition, phosphorylation levels of p38 and ERK1/2 induced by insulin and T3 were obviously lower than those induced by Ang II. The lower phosphorylation levels induced by insulin and T3 may not significantly change *RBM20* transcriptional activity.

Collectively, this study identified underlying molecular mechanisms of external stimuli-mediated splicing via RBM20. These results revealed that insulin, T3, and Ang II regulate titin pre-mRNA splicing in an RBM20-dependent manner. In addition, external stimuli increased mRNA levels of *RBM20* through their enhancement of *RBM20* promoter transcriptional activity. Additionally, Ang II can activate the MAPK/ELK1 signaling pathway and thus trigger the *RBM20* transcriptional activity (Figure 6). The results of this study may serve as a guide for the development of treatment for heart diseases by regulation of pre-mRNA splicing.

## 4. Materials and Methods

### 4.1. Experimental Animals and Tissue Samples

Wild type (WT) and *RBM20* knockout (KO) rats used in this study were crosses of Sprague-Dawley (SD) × Brown Norway (BN). All strains were originally obtained from Harlan Sprague Dawley, Indianapolis, IN, USA. KO rats are derived from a spontaneous mutant. Animals were fed standard rodent chow. The animal protocols used in this work were evaluated and approved by the Institutional Animal Use and Care Committee of the University of Wyoming (protocols: 20130920WG0027, 19 September, 2015; 20180206BC00293, 5 February 2019). Hearts of 1-day, 3- month, and 6-month in WT and KO rats were collected. Procedures of STZ, T3, PTU, and Ang II treatments were the same as described in previous publications [18,19]. After treatments, heart tissues were harvested and snap-frozen in liquid nitrogen immediately after removal and stored at −80 °C until use.

### 4.2. Neonatal Rat Cardiomyocytes (NRCMs) Isolation and Treatment

Primary cardiomyocytes were isolated from the heart tissue of neonatal WT rats (0–7 day) using the NRCM isolation system [18,19]. After NRCMs were plated for 24 h, the culture medium was switched into serum-free medium overnight. In different treatments, the serum-free medium was supplemented with insulin (175 nM), T3 (10 nM), and Ang II (100 nM), respectively. NRCMs cultured with serum-free medium without any supplementary were utilized as negative control. Each group had three replicates.

### 4.3. Reverse Transcriptional PCR (RT-PCR) and Quantitative Real-Time PCR (qPCR) Analysis

Total RNAs from heart tissues described above were extracted with Trizol (Invitrogen, Carlsbad, CA, USA) according to the manufacturer’s instruction. Total RNAs were treated with DNase I (Invitrogen, Carlsbad, CA, USA) before cDNA synthesis using the ImProm-II Reverse Transcription System (Promega, Madison, WI, USA) with random primers (Invitrogen, Carlsbad, CA, USA). Primers used in RT-PCR to amplify TTN Z-band (exon 7–10 and exon 10–14), I-band (exon 54–56 and exon 66–68), and M-band (exon 361–363) were the same as previously published [12,14] (Appendix A). Thirty cycles were optimized for ratio calculation of different variants. PCR products were analyzed by 2% DNA agarose gel electrophoresis. DNA gel was stained with ethidium bromide and visualized under UV light. ChemiDoc Imaging System was used to capture gel images. DNA band density ratio was quantified with NIH ImageJ. NRCMs treated for 48 h were collected for total RNA extraction, cDNA synthesis, and qPCR analysis. A qPCR was performed using a Bio-Rad CFX96 Real-Time Detection System and SYBR Green PCR Master Mix. Rat Glyceraldehyde-3-phosphate (GAPDH) gene was chosen as an endogenous control. All reactions were carried out in triplicate. Relative expression ratios were calculated with the formula 2^−ΔΔCt^ as described in a previous publication [44].

### 4.4. Western Blot Analysis

NRCMs treated after 5–10 min were collected for western blot analysis to detect the hallmark proteins in the MAPK signaling cascade. NRCMs were homogenized in urea-thiourea buffer (8 M urea, 2 M thiourea, 75 mM DTT, 3% SDS, 0.05% bromophenol blue, 0.05 M Tris, pH = 6.8) as described previously [20]. Total protein was separated by SDS-PAGE gel and transferred onto PVDF membrane. The membrane was probed with antibodies against p38, phospho-p38 (Thr180/Tyr182), ERK1/2, phospho-ERK1/2 (Thr202/Tyr204), JNK, and phospho-JNK (Thr183/Tyr185). Secondary antibodies are goat anti-mouse IgG- and goat anti-rabbit IgG-conjugated with horseradish peroxidase. Anti-GAPDH was used as protein loading control. Western blot was developed with Enhanced Chemiluminescence (ECL) western blotting substrate and exposed to CL-Xposure film. Protein band density was quantified with NIH ImageJ. Each group has three replicates.

### 4.5. Bioinformatics Analysis

Putative transcription factor-binding sites were predicted using the following online tools: MatInspector (http://www.genomatix.de/), JASPAR database (http://hfaistos.uio.no:8000/) [45], and Promo (http://alggen.lsi.upc.es/cgi-bin/promo_v3/promo/promoinit.cgi?dirDB = TF_8.3) [46,47,48]. Multiple sequence alignment was performed using BioXM 2.6.

### 4.6. Dual-Luciferase Activity Assay

Genomic fragment (−1120 bp ~ +120 bp) containing promoter of mouse *RBM20* gene (Genebank accession number: NC_000085.6) was amplified from mouse genomic DNA which was extracted from toes using DirectPCR Lysis Reagent and constructed into pGL3-basic vector (CON). PCR performed with c-jun binding site deletion in forward primer and overlap PCR was used to introduce site-directed deletion of c-jun (BS1) and ELK1 binding site (BS2) in *RBM20* promoter region, and then the fragment was constructed into pGL3-basic vector (MUT). Primers used to construct report vector were listed in Appendix A. H9C2 cells were seeded into 48-well plate. CON and MUT vectors were transfected by lipofectamine 2000 according to its instruction. Twelve hours after transfection, the culture medium was switched into serum-free medium overnight. Then, the serum-free medium was supplemented with insulin (175 nM), T3 (10 nM), and Ang II (100 nM), respectively. Cells cultured with serum-free medium without any supplementary were used as a negative control. Each group had three replicates. After 24 h, cells were collected and luciferase activities were measured using Dual-Luciferase Report Assay System. Relative luciferase activity was calculated as the ratio of the signal of firefly to Renilla luciferase.

### 4.7. Statistics

Results were analyzed using SPSS 20.0 and reported as mean ± standard error (SE). Statistical analysis was performed using Student’s *t*-test when compared between two groups. Chi-square was used to detect the significance of splicing patterns between different groups. *p* < 0.05 was considered statistically significant (* *p* < 0.05, ** *p* < 0.01, *** *p* < 0.001).

## Figures and Tables

**Figure 1 ijms-20-05059-f001:**
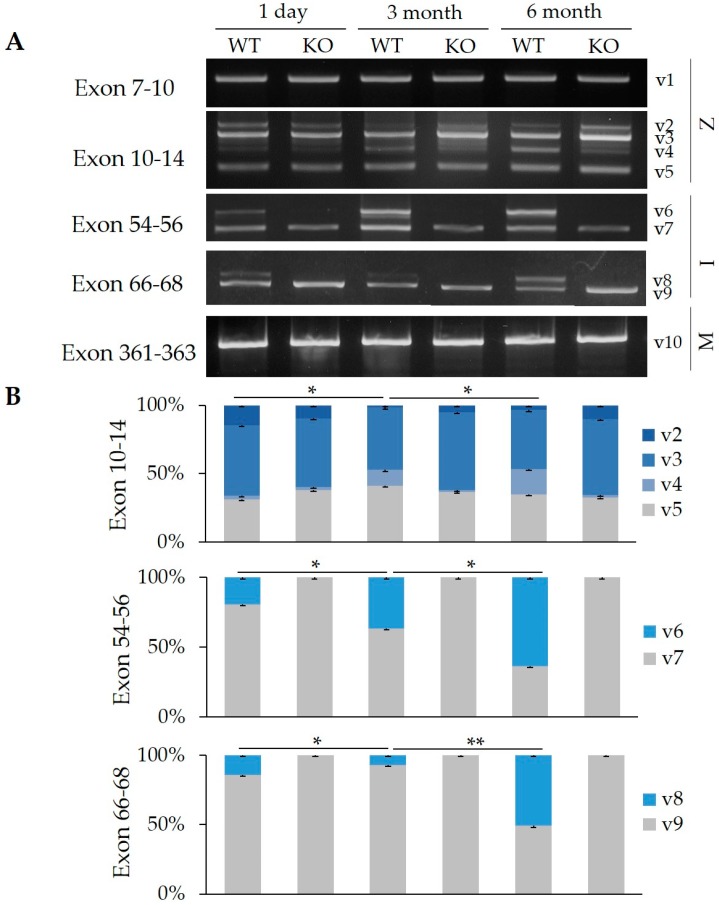
Splicing of Z-, I-, and M-band titin during development. (**A**) RT-PCR results with primers spanning exons 7–10 and exons 10–14 in the Z-band, exons 54–56 and exons 66–68 in the I-band, and exons 361–363 in the M-band. (**B**) Quantification of band density presented for different variants amplified by primers spanning exons 10–14, exons 54–56, and exons 66–68. Significant differences in v4, v6, and v7 ratios were compared in exons 10–14, 54–56, and 66–68 regions, respectively. Mean ± SE (*n* = 3), * *p* < 0.05, ** *p* < 0.01. v, variant; 1–10, different variants. WT, wild type; KO, RNA binding protein 20 (*RBM20*) knockout. Z, Z-band. I, I-band. M, M-band.

**Figure 2 ijms-20-05059-f002:**
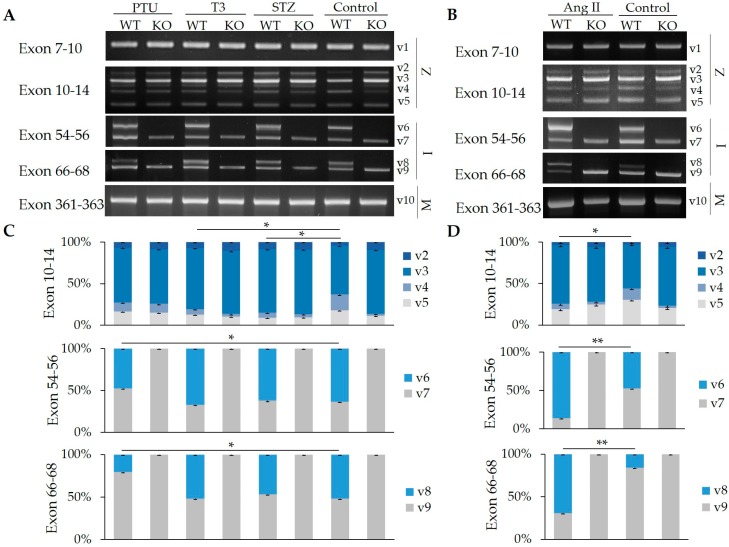
Z-, I-, and M-band titin splicing in WT and KO rats treated with propylthiouracil (PTU), triiodothyronine (T3), streptozotocin (STZ), and angiotensin II (Ang II). (**A**) RT-PCR results of Z-, I-, and M-band titin splicing with PTU, T3, and STZ; (**B**) RT-PCR results of Z-, I-, and M-band titin splicing with Ang II treatment. (**C**) Quantification of band density of different splicing variants amplified by primers spanning exons 10–14, exons 54–56, and exons 66–68 with PTU, T3, and STZ; (**D**) Quantification of band density of splicing variants amplified by primers spanning exons 10–14, exons 54–56, and exons 66–68 with Ang II treatment. Mean ± SE (*n* = 3), * *p* < 0.05, ** *p* < 0.01. v, variants; 1–10, different variants; WT, wild type; KO, *RBM20* knockout; Z, Z-band. I, I-band. M, M-band.

**Figure 3 ijms-20-05059-f003:**
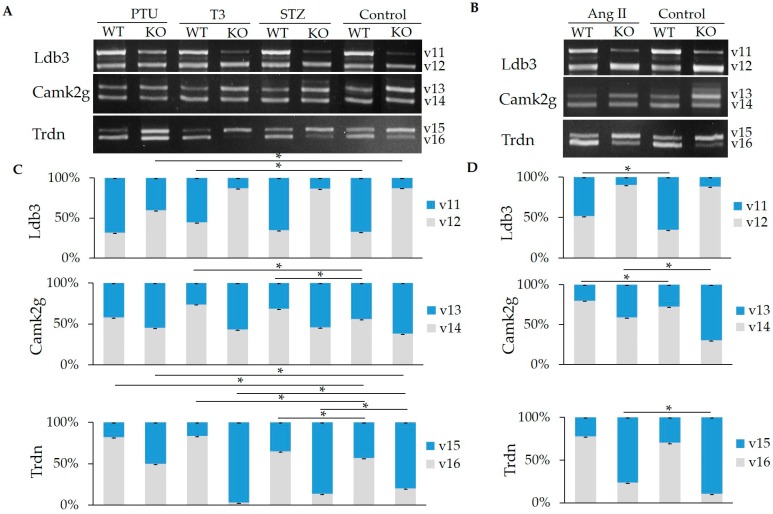
Splicing pattern of *Ldb3*, *Camk2g*, and *Trdn* genes in WT and KO rats treated with PTU, T3, STZ, and Ang II. (**A**) RT-PCR results of *Ldb3*, *Camk2g*, and *Trdn* splicing under PTU, T3, and STZ treatment in heart tissues; (**B**) RT-PCR results of *Ldb3*, *Camk2g*, and *Trdn* splicing under Ang II treatment in heart tissues. (**C**) Quantification of band density of different splicing variants of *Ldb3*, *Camk2g*, and *Trdn* under PTU, T3, and STZ treatment; (**D**) Quantification of band density of different splicing variants of *Ldb3*, *Camk2g*, and *Trdn* under Ang II treatment. v, variants; 11–16, different variants. Mean ± SE (*n* = 3), * *p* < 0.05.

**Figure 4 ijms-20-05059-f004:**
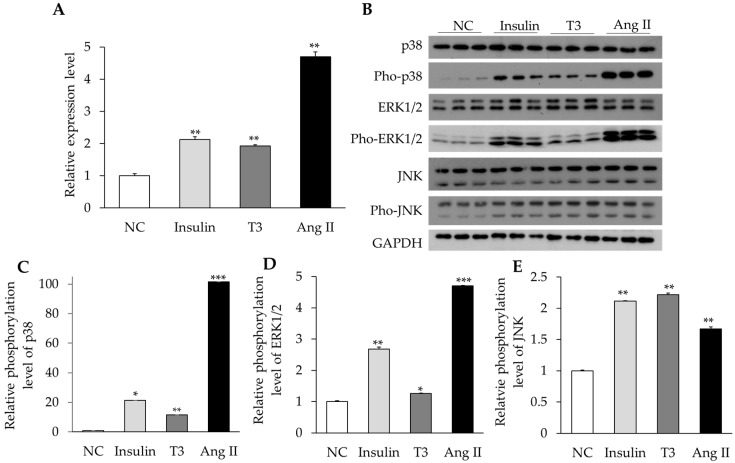
*RBM20* transcriptional activity and activation of the mitogen-activated protein kinase (MAPK) signaling through insulin, T3, and Ang II. (**A**) Relative mRNA levels of *RBM20* under different treatments; (**B**) Western blot analysis of hallmark proteins in the MAPK signaling cascade; (**C**–**E**) Quantification of relative phosphorylation levels of p38, ERK1/2, and JNK in comparison to their individual pho-antibodies. Glyceraldehyde-3-phosphate (GAPDH), protein loading control; NC, negative control; Mean ± SE (*n* = 3), * *p* < 0.05, ** *p* < 0.01, *** *p* < 0.001. Statistical significance was compared between different treatments and their individual negative controls.

**Figure 5 ijms-20-05059-f005:**
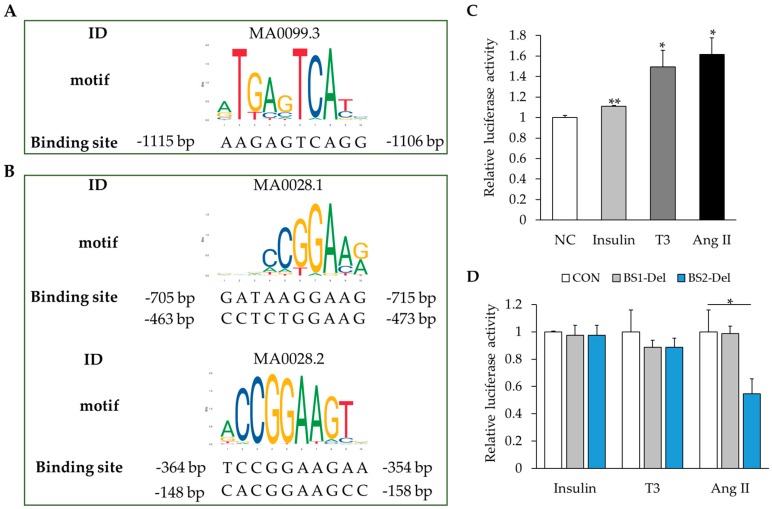
Analysis of promoter activity of *RBM20* gene. (**A**) The binding site of c-jun in the *RBM20* promoter region; (**B**) Binding sites of ELK1 in the *RBM20* promoter region; (**C**) Relative luciferase activities of CON *RBM20* promoter with different treatments. Luciferase activity of untreated control was set as “1.0”; (**D**) Relative luciferase activities of MUT *RBM20* promoter with different treatments. With each treatment, the luciferase activity of the CON promoter was set as “1.0”. CON, pGL3-basic vector containing *RBM20* promoter fragment. BS1-Del, pGL3-basic vector containing *RBM20* promoter fragment from which binding site of c-jun was deleted. BS2-Del, pGL3-basic vector containing RBM20 promoter fragment from which binding sites of ELK1 were deleted. Mean ± SE (*n* = 3), * *p* < 0.01, ** *p* < 0.01.

**Figure 6 ijms-20-05059-f006:**
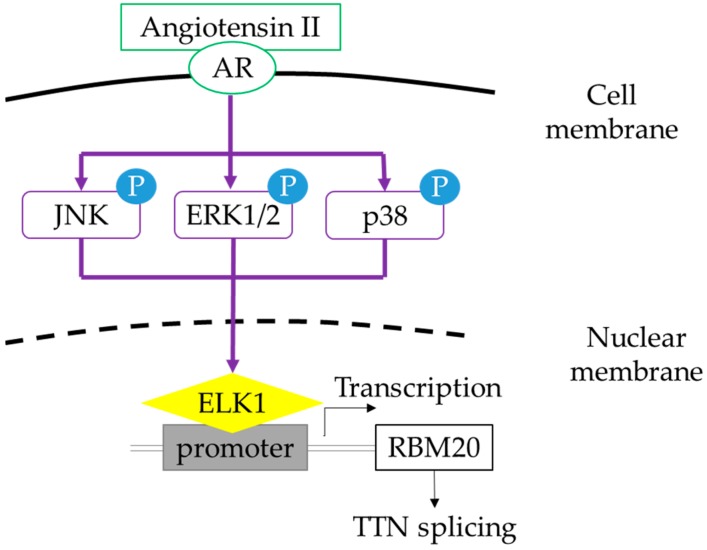
Schematic diagram depicting the MAPK/ELK1 signaling pathway linking Ang II-RBM20-titin splicing. Ang II activates and phosphorylates three signal molecules, JNK, ERK1/2, and p38 in the MAPK signaling pathway, and then activates the expression of ELK1. ELK1 regulates the transcriptional and mRNA expression of *RBM20*, which regulates pre-mRNA splicing of titin. AR, Angiotensin II receptor.

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
