# Peer review of "Angiotensin II Influences Pre-mRNA Splicing Regulation by Enhancing RBM20 Transcription Through Activation of the MAPK/ELK1 Signaling Pathway"

_ijms, 2019, doi:10.3390/ijms20205059_

Round 1

Reviewer 1 Report

The authors explore the effects of various hormone treatments on alternate splicing events in titin mRNA gene splicing, especially with respect to the role of the RBM20 splicing factor.  This is of general importance as the sarcomeric protein titin and other genes regulated at the splicing level by RBM20 play important roles in normal cardiac function and have been implicated in cardiomyopathies.

The authors first examine the role of RBM20 in titin mRNA splicing by examining RNA taken from hearts from rats aged 1 day, 3 months, and 6 months, both with and without an RBM20 knockout.  Analysis was performed using semi-quantitative RT-PCR and agarose gel analysis of cDNA fragments representing the Z-, I-, and M-band regions of the alternately spliced titin mRNA.  Results suggest that inactivation of RMB20 affects the relative frequencies of alternate splicing events in the Z-band region of 3 and 6 month old hearts, and the I-band region of all age rat hearts.  Though statistical analysis is not reported for the experiments shown in figure 1, the authors claim in lines 86-88 that RBM20 primarily regulates splicing primarily in the I-band.  It is not clear why Z-band effects are not included in this conclusion.  Though the semi-quantitative nature of end-point RT-PCR allows only general indications of changes in levels of alternate splicing events, these observations suggest one of several possibilities: 1) RBM20 plays different roles in alternate splicing regulation in different regions of the titin mRNA at different ages, or 2) different regions of the titin mRNA have different sensitivities to regulators of splicing fidelity in the absence of RBM20 and are therefore more prone to splicing errors, or 3) alternate mRNA splice products have different stabilities in the absence of RBM20 for some reason.  The alternate splice products that do not retain an open reading frame are not identified, so we do not know which splice variants would be subject to nonsense-mediated mRNA decay.  All of these possibilities should be discussed.   

The authors then test the idea that hormones affect alternate splicing patterns of titin through RBM20 activity by repeating the frequency/spectrum analysis of alternate splicing events in wild type and RBM20 knockout rats in the presence of PTU, T3, STZ, and Ang II.  While in general there were greater differences between controls and knockouts, there were some effects on alternate splicing events associated with some hormone treatments.  T3, STZ, and Ang II treatments phenocopy RBM20 inactivation, suggesting these reagents interfere with a normal function of RBM20 or act in a separate antagonistic pathway.  In the I-band regions, PTU and Ang II treatments affect splicing patterns, but neither treatment phenocopies RBM20 inactivation, which masks the effects of hormone level changes.  This suggests that the hormones play a minor role in RBM20 activity, or that RBM20 plays a separate role that is required for any alternate splice patterns to be seen.  It is a concern that the controls for the Ang II and PTU experiments do not look similar in the I-band region. 

To test the generality of the hormonal effects on genes associated with heart disease, the authors repeated RT-PCR analysis of alternate splicing events in Ldb3, Camk2g, and Trdn mRNAs.  Alternate splice variant accumulation of Ldb3 is affected by T3 and Ang II.  However, the presence of PTU seems to lessen the effects of inactivating RBM20, suggesting inhibition of endogenous thyroid hormone may work in partly redundant pathways.  Likewise, splicing patterns of Camk2g are affected by the presence of T3, STZ, and Ang II.  In this case, the effects of Ang II appear to partly lessen the effects of RBM20 inactivation, again suggesting they contribute additively.  Splicing patterns of Trdn mRNAs are also affected by PTU, T3, and STZ.  In this case, the effects of Ang II, and PTU lessen the effects of RBM20 inactivation, while the effects of T3 and STZ enhance the effects.  These suggest parallel antagonistic and cooperative pathways, respectively.

It has been shown previously that various hormone treatments stimulate the pI3K/AKT/mTOR pathway.  Here the authors show that insulin, T3, and Ang II also stimulate the MAPK pathway by promoting phosphorylation of p38, ERK1/2, and JNK.  To determine whether this pathway is responsible for promoting RBM20 activity at the transcriptional level, the authors use luciferase assays to test the RBM20 promoter response to hormone treatments.  As Figure 2 shows, T3, Ang II, and inhibition of insulin all phenocopy RBM20 KO in region Z, suggesting T3 and Ang II are antagonistic to RBM20 activity, while insulin acts cooperatively.  It is therefore somewhat surprising that all three promote transcription of RBM20.  This suggests the effects of altered hormone environments may have complex effects on alternate splicing of titin mRNA, not all of which are reflected in RBM20 transcription rates.  Mutations in putative c-jun and ELK binding sites reveal that the ELK1 binding motif is necessary to prevent full transcriptional activation of the RBM20 promoter by Ang II.  This does not demonstrate directly that ELK1 is bound to that site or plays a role in RBM20 expression.  Nor is it demonstrated directly that increased levels of RBM20 mRNA levels play a significant role in RBM20 protein levels or activity.      

This report contains useful information but overstates several claims.  Corrections should be made as follows:

While it is known that RBM20 is a splicing factor, it cannot be assumed that all changes in splicing patterns seen in RBM20 KOs result from loss of RBM20 activity at the exons being examined.  Loss of RBM20 could have indirect effects on splice variant stability or non-regulated loss of splicing fidelity.  Text should be included to address these formal possibilities. Statistical analysis should be provided for the experiments in Fig. 1. The title and statements presented in the text suggest that ELK1-mediated regulation of RBM20 transcription plays a role in splicing regulation of titin. The text should be modified to include several alternate explanations of the data that were not tested.  It has not been demonstrated directly that ELK1 binds to the candidate ELK1 binding motif in the RBM20 promoter, nor has it been demonstrated that increased levels of RBM20 mRNA is either necessary or sufficient for the observed effects on titin alternate splicing patterns.    

Minor points: the stray numbers over the bars in Figures 2 and 3 need to be removed, and the p values assigned to the asterisks in the figure 5 legend needs to be corrected.  The text needs to be edited for clarity and conventional English. 

Author Response

Reviewer 1

…Though statistical analysis is not reported for the experiments shown in figure 1, the authors claim in lines 86-88 that RBM20 primarily regulates splicing primarily in the I-band.  It is not clear why Z-band effects are not included in this conclusion.  Though the semi-quantitative nature of end-point RT-PCR allows only general indications of changes in levels of alternate splicing events, these observations suggest one of several possibilities: 1) RBM20 plays different roles in alternate splicing regulation in different regions of the titin mRNA at different ages, or 2) different regions of the titin mRNA have different sensitivities to regulators of splicing fidelity in the absence of RBM20 and are therefore more prone to splicing errors, or 3) alternate mRNA splice products have different stabilities in the absence of RBM20 for some reason.  The alternate splice products that do not retain an open reading frame are not identified, so we do not know which splice variants would be subject to nonsense-mediated mRNA decay.  All of these possibilities should be discussed.

Response:Thanks the reviewer for pointing out the missed statistical analysis and giving constructive suggestions on Z-band splicing. We added the statistical analysis in figure 1 and added the discussion regarding the splicing pattern and expression level of Z-band variants in the main text.

… This suggests that the hormones play a minor role in RBM20 activity, or that RBM20 plays a separate role that is required for any alternate splice patterns to be seen.  It is a concern that the controls for the Ang II and PTU experiments do not look similar in the I-band region. 

Response:We disagree that hormones play a minor role in RBM20 activity as reasoned above. Controls for PTU and Ang II experiments has the same splicing pattern, but different expression level of the variants in the I-band region. The reason is that we treated the animals at 3-month old with Ang II, but at 6-month old with PTU, so the control used for PTU and Ang II are 6 and 3 month old respectively. If you look at the splicing pattern and expression level of these variants in the development study indicated in figure 1, you will see that these two controls are consistent with development. The age of the control samples used in PTU and Ang II has been added to the main text.

…As Figure 2 shows, T3, Ang II, and inhibition of insulin all phenocopy RBM20 KO in region Z, suggesting T3 and Ang II are antagonistic to RBM20 activity, while insulin acts cooperatively.  It is therefore somewhat surprising that all three promote transcription of RBM20.  This suggests the effects of altered hormone environments may have complex effects on alternate splicing of titin mRNA, not all of which are reflected in RBM20 transcription rates.  Mutations in putative c-jun and ELK binding sites reveal that the ELK1 binding motif is necessary to prevent full transcriptional activation of the RBM20 promoter by Ang II.  This does not demonstrate directly that ELK1 is bound to that site or plays a role in RBM20 expression.  Nor is it demonstrated directly that increased levels of RBM20 mRNA levels play a significant role in RBM20 protein levels or activity.

Response:Although the hormones phenocopy RBM20 KO in region Z, it does not suggest the hormones T3 and Ang II are antagonistic to RBM20 activity. It could be that RBM20 does not have binding activity to exons or introns in region Z. Even though these hormones still promote RBM20 activity, they don’t enable the splicing in region Z in either presence or absence of RBM20. Therefore, in our opinion, promoting transcription of RBM20 by all three hormones is not necessary to change splicing pattern in region Z. We agree with the reviewer that we didn’t provide the direct evidence of ELK1 binding sites to RBM20 and effect of ELK1 on protein changes of RBM20 in current article. Our future study will use CHIP assay to identify the direct binding sites of ELK1 to the promoter of RBM20 and also detect if ELK1 impacts on RBM20 protein level.

This report contains useful information but overstates several claims.  Corrections should be made as follows: While it is known that RBM20 is a splicing factor, it cannot be assumed that all changes in splicing patterns seen in RBM20 KOs result from loss of RBM20 activity at the exons being examined.  Loss of RBM20 could have indirect effects on splice variant stability or non-regulated loss of splicing fidelity.  Text should be included to address these formal possibilities. Statistical analysis should be provided for the experiments in Fig. 1.

Response:Thank the reviewer for the constructive suggestions. We agree and have added the statement in the main text. Also, the statistical analysis has been provided in Fig. 1.

The title and statements presented in the text suggest that ELK1-mediated regulation of RBM20 transcription plays a role in splicing regulation of titin. The text should be modified to include several alternate explanations of the data that were not tested. It has not been demonstrated directly that ELK1 binds to the candidate ELK1 binding motif in the RBM20 promoter, nor has it been demonstrated that increased levels of RBM20 mRNA is either necessary or sufficient for the observed effects on titin alternate splicing patterns.    

Response:Previous studies indicate that insulin, T3, and Ang II can increase the phosphorylation level of ELK1. Luciferase activity detection assay with site-directly deletion was used as direct evidence to detect the effect of transcriptional factor on gene promoter (Yao et al., 2016, J. Cell. Physiol.). However, we agree that we didn’t provide direct evidence for the ELK1 binding sites on the promoter of RBM20 which will be done in our future study using CHIP assay. We have added the alternative explanations in the main text. In addition, our previous studies have shown that titin splicing is RBM20-dependent, but we agree that it is unknown if the expression level is sufficient for the observed effects on titin splicing, which is discussed in the main text.

Minor points: the stray numbers over the bars in Figures 2 and 3 need to be removed, and the p values assigned to the asterisks in the figure 5 legend needs to be corrected.  The text needs to be edited for clarity and conventional English. 

Response:Thanks the reviewer for pointing this out. We have corrected the errors and made changes.

Reviewer 2

the methods are in general well described. However, I miss any remark on treating the RT-PCRs in the way these could serve for quantifications. As the PCR should be stopped at the exponential phase to be comprehended as a semi-quantitative method, the authors shall provide the readers with some details on this method.

Response:This study was designed to quantify the ratio of internal bands (variants in individual lanes), so it is not important to stop the PCR amplification at exponential phase. However, to remove the effect of amplification efficiency of each variant (may cause difference if reaching exponential phase), we optimized the PCR condition through amplification cycles to make sure we can see all clear bands on the gel, but not reaching exponential phase. PCR cycles of 30 was the optimized condition for this study. We have added this remark to the method.

in the bar charts, the standard error is shown. Is there any special reason for that instead of using the standard deviation? I believe the usage of standard deviation is more common and usually more appropriate.

Response:Standard deviation measures the amount of variability or dispersion for a subject set of data from the mean, while the standard error of the mean measures how far the sample mean of the data is likely to be from the true population mean. In our research, three independent experiments were performed (n=3), and statistical significance of mean was compared between different groups, so we believe it is appropriate for us to use standard error in our bar chart. Here are examples about the use of standard error bar from other researchers in their bar chart (Liu, et al. 2016, Cardiovascular research, 111(1), 56-65; Han, P., et al. 2014, 514(7520), 102). However, we believe some other researchers may use standard deviation to present their results.

maybe the difference between the treatment of PTU, T3 and STZ contra AngII could be mentioned. At least the Ang II treatment results are always shown separately and their controls seem to be different from that of PTU, T3 and STZ treatment (e.g. in Fig.3D-C, respectively).

Response:This has been addressed in the reviewer 1’s point 2. In addition, due to the gel well limitation, we ran these samples in different gels.

the promoter fragment of RBM20 used for luciferase analyses should be specified, since the results of luc. analyses are often connected to the particular promoter stretch.

Response:The localization of the promoter fragment of RBM20 has been added in the method.

some of the concluding statements seem not to be in total concordance with the data, at least as I understand it. Excuse me if I missed something. The examples of these unclarities: lines 86-87: Collectively, these results suggest that RBM20 does not regulate titin splicing patterns in both Z-band and M-band, … - this is not concordant with Figure 1 that clearly shows the changes in titin exons 10-14 (Z-band) alternative splicing during development in the wt rats but no changes in the KO rats. I could understand that these differences might not reach the statistical significants, but in that case, I would choose other words.

Response:The variant differences in WT have been discussed as addressed in the reviewer 1’s point 1, and we have modified the sentence with different wording.

lines 223-225: but only Ang II can trigger transcriptional factor ELK1-bound promoter of RBM20 for its transcriptional activity – as I understand it, several factors (ins., T3 and Ang II) increased the transcription from the RBM20 promoter, but probably only Ang II in the ELK-1 dependent manner. Maybe it is just an unclear statement.

Response:We have re-stated this sentence “but among tested hormones, only Ang II can activate transcriptional factor ELK1-bound promoter of RBM20…”

lines 254-255: This study indicated that titin Z- and M-band splicing does not experience developmental changes… - this does not correspond to the results shown in Figure 1 (v 4), the Z-band seems to be developmentally regulated. For the same reason, I do not understand another statement (lines 256-257): We also found that titin Z- and M-band splicing is not regulated by RBM20 … 

Response:We have re-stated these sentences “This study indicated that the Z-band variant v4 experiences developmental changes, but not other variants (v2, 3 and 5). The M-band variant does not experience developmental changes. I-band titin variants are drastically regulated developmentally. We also found that RBM20 is not a major regulator for the Z- and M-band titin splicing…”

some statements are not totally clear to me, e.g.: lines 126-128: Taken together, although hormones can alter the expression level of some variants in titin Z- and I-band in the WT group, but splicing pattern has no changes in all treated groups.

Response:We have re-stated this sentence “To summarize, although hormone treatments can alter the expression level of some variants in Z- and I-band titin in WT group, splicing pattern was not altered. Splicing pattern differences between WT and KO hearts were only observed in the I-band.”

The title of the article seems to be confounding to me: „Angiotensin II enhances RBM20 Transcriptional Activity in the Regulation of Pre-mRNA Splicing...“ sound as if the transcriptional activity of RBM20 itself was important for the splicing regulation. For me, as I understand it, it would be clearer as e.g.: „Angiotensin II enhances RBM20 Transcription, influencing (affecting) Pre-mRNA Splicing Regulation …“

Response:We agree with the reviewer and changed the title to “Angiotensin II Influences Pre-mRNA Splicing regulation by Enhancing RBM20 Transcription Through Activation of the MAPK/ELK1 Signaling Pathway”

The way of indicating statistical significance in the charts is non-standard. It can be comprehensible, just the legends may be adapted accordingly. 

Response:We have made changes accordingly.

The clarity of figures would benefit from less text around. E.g. in figure 3, the extensive labels such as „Percentage of Trdn variants “is unnecessary. The percents are clear from the charts, the variants are denoted at the opposite side of the chart. So „Trdn“ could be enough. On the other hand, in Figure 4A, the chart could be more labeled. The Y-axis lacks the note that the relative expression pertains to RBM20. 

Response:We agree with the reviewer and made changes in these figures.

The results seem to be a little bit longish in some parts. Maybe it is not necessary to describe everything that is shown in the charts including the negative results. If some of these were skipped, the important rest would become more apparent. The reader can find the rest in the charts.

Response:Thanks for the reviewer’s suggestion. We have made changes.

A simple figure of splicing variants that were tested would be beneficial. E.g. as a supplement.

Response:The schematic diagram of the splicing variants of Z-, I-, and M-band titin has been reported by our group previously. We have cited the publications in this manuscript which the readers can refer to.

Reviewer 2 Report

The authors provided an interesting study on the regulation of several heart muscle genes splicing by hormones such as insulin, triiodothyronine and angiotensin II. Primarily, they studied alternative splicing regulation of titin by RMB20 in developing rat hearts, showing that the main developmental splicing changes occur in the I-band of the gene. Similarly, the I-band showed the highest response after the hormonal treatment. Other heart muscle genes showed some alternative splicing changes upon the treatment, but these were not consistent and easily explicable. Using western blotting, the authors showed that the studied hormonal response might employ the MAPK pathway. Finally, the authors proved the importance of RBM20 factor in the hormone response by using a dual-luciferase assay system. With this method, they showed the increased transcription from the RBM20 promoter upon the hormonal treatment. Yet the effect of the MAPK pathway transcription factor, ELK-1, on RBM20 promoter activity was only detected with angiotensin II treatment. 

The presented piece of work describes interesting research conducted on an important subject. Results of this and future projects could eventually lead to the development of new therapies for serious cardiac conditions. The manuscript is well structured, overall clearly written, with lucid figures. Yet some of the statements are unclear and/or inexact, as specified below.

Major comments:

- the methods are in general well described. However, I miss any remark on treating the RT-PCRs in the way these could serve for quantifications. As the PCR should be stopped at the exponential phase to be comprehended as a semi-quantitative method, the authors shall provide the readers with some details on this method.

- in the bar charts, the standard error is shown. Is there any special reason for that instead of using the standard deviation? I believe the usage of standard deviation is more common and usually more appropriate.

- maybe the difference between the treatment of PTU, T3 and STZ contra AngII could be mentioned. At least the Ang II treatment results are always shown separately and their controls seem to be different from that of PTU, T3 and STZ treatment (e.g. in Fig.3D-C, respectively).

- the promoter fragment of RBM20 used for luciferase analyses should be specified, since the results of luc. analyses are often connected to the particular promoter stretch.

- some of the concluding statements seem not to be in total concordance with the data, at least as I understand it. Excuse me if I missed something. The examples of these unclarities:

- - lines 86-87: Collectively, these results suggest that RBM20 does not regulate titin splicing patterns in both Z-band and M-band, … - this is not concordant with Figure 1 that clearly shows the changes in titin exons 10-14 (Z-band) alternative splicing during development in the wt rats but no changes in the KO rats. I could understand that these differences might not reach the statistical significants, but in that case, I would choose other words.

- - lines 223-225: but only Ang II can trigger transcriptional factor ELK1-bound promoter of RBM20 for its transcriptional activity – as I understand it, several factors (ins., T3 and Ang II) increased the transcription from the RBM20 promoter, but probably only Ang II in the ELK-1 dependent manner. Maybe it is just an unclear statement.

- - lines 254-255: This study indicated that titin Z- and M-band splicing does not experience developmental changes… - this does not correspond to the results shown in Figure 1 (v 4), the Z-band seems to be developmentally regulated. For the same reason, I do not understand another statement (lines 256-257): We also found that titin Z- and M-band splicing is not regulated by RBM20 … 

- some statements are not totally clear to me, e.g.:

- - lines 126-128: Taken together, although hormones can alter the expression level of some variants in titin Z- and I-band in the WT group, but splicing pattern has no changes in all treated groups.

Minor comments:

- The title of the article seems to be confounding to me: „Angiotensin II enhances RBM20 Transcriptional Activity in the Regulation of Pre-mRNA Splicing...“ sound as if the transcriptional activity of RBM20 itself was important for the splicing regulation. For me, as I understand it, it would be clearer as e.g.: „Angiotensin II enhances RBM20 Transcription, influencing (affecting) Pre-mRNA Splicing Regulation …“

- The way of indicating statistical significance in the charts is non-standard. It can be comprehensible, just the legends may be adapted accordingly. 

- The clarity of figures would benefit from less text around. E.g. in figure 3, the extensive labels such as „Percentage of Trdn variants“ is unnecessary. The percents are clear from the charts, the variants are denoted at the opposite side of the chart. So „Trdn“ could be enough.

- On the other hand, in Figure 4A, the chart could be more labeled. The Y-axis lacks the note that the relative expression pertains to RBM20. 

- The results seem to be a little bit longish in some parts. Maybe it is not necessary to describe everything that is shown in the charts including the negative results. If some of these were skipped, the important rest would become more apparent. The reader can find the rest in the charts.

- A simple figure of splicing variants that were tested would be beneficial. E.g. as a supplement.

Author Response

(The authors gave the same response as above.)
